

# Impact of cooling on shaping ability of thermally treated files in canal models with double curvature

Ahmed Jamleh[1,2], Hajar Albanyan[1,2], Ali Alaqla[1,2], Hamad Alissa[1,2], Nawaf Alshetan[3], Sulaiman Alraffa[3] and Abdulmohsen Alfadley[1,2]

[1] Department of Restorative and Prosthetic Dental Sciences, College of Dentistry, King Saud bin Abdulaziz University for Health Sciences, Riyadh, Saudi Arabia
[2] King Abdullah International Medical Research Center, Riyadh, Saudi Arabia
[3] Dr. Sulaiman Al-Habib Medical Group, Riyadh, Saudi Arabia

## ABSTRACT

**Background**. This study compared the ability of thermally treated files in shaping simulated canals with double curvature. Fifty-six canals were enlarged to a final size of 25 with ProTaper Next (PTN) or ZenFlex (ZF).

**Materials**. Half of the samples were shaped with cooled files ($n = 14$ each). The amount of removed resin was measured and canal deviation was determined at eight levels. Shaping time and maximum shaping torque values were also recorded. Data were statistically analyzed using analysis of variance and LSD, Kruskal–Wallis, and chi-square tests at a 0.05 significance level.

**Results**. Compared to PTN and cooled PTN, ZF and cooled ZF required lesser time to shape the canals. The maximum torques were found comparable between the groups. All the groups generated negligible deviations at every canal level evaluated and maintained the canal geometry. Although not significant, the cooled PTN and ZF files exhibited lesser canal deviations than their counterparts.

**Conclusion**. All groups demonstrated similar shaping ability whilst maintaining the original curvature of the canal in simulated canals with double curvature. However, ZF groups were able to shape the canals faster than PTN groups. There was a trend that cooled files made lesser canal deviations compared to their counterparts.

## INTRODUCTION

Canal shaping is performed to remove pulpal tissue and make a continuously tapered space that facilitates irrigation, placement of medicament, and obturation in three dimensions (*Hülsmann, Peters & Dummer, 2005*). It is highly prevalent to find canals with multiple curves in different planes. This is challenging as it increases the incidence of endodontic mishaps and canal transportation as endodontic files tend to straighten the root canal with their cutting effect toward the inner aspect of the canal curvature and toward the outer aspect of the curvature at the canal exit (*Hülsmann, Peters & Dummer, 2005*; *Peters, 2004*). These mishaps may weaken the root structure and negatively affect canal disinfection and obturation (*Hülsmann, Peters & Dummer, 2005*; *Peters, 2004*).

Corresponding author
Ahmed Jamleh, aojamleh@gmail.com

It is evident that nickel-titanium (NiTi) files are able to keep the original canal shape without significant mishaps especially in curved canals due to their superelastic behavior and shape-memory properties (*Hülsmann, Peters & Dummer, 2005*; *Peters, 2004*). The properties are derived from the phase transformation between two crystalline structures, austenite and martensite. The NiTi alloy is stiff and hard when it is in the austenite phase and is flexible and soft when it is in the martensite phase (*Jamleh et al., 2012*; *Shen et al., 2013*). Over the three decades of NiTi rotary file development and clinical use, advancements in many aspects have been made to improve the file performance such as cross-section design, tip, taper, cutting blades, alloys, heat treatment, and motion kinematics (*Shen et al., 2013*; *Zhang, Cheung & Zheng, 2010*; *Lopes et al., 2013*; *Elnaghy & Elsaka, 2014*; *Bürklein, Mathey & Schäfer, 2015*; *Kyaw Moe et al., 2018*; *Zupanc, Vahdat-Pajouh & Schäfer, 2018*).

The ProTaper Next system (PTN) (Dentsply Sirona, Ballaigues, Switzerland) has an asymmetrical rectangular cross-section with two cutting edges with a variable taper and is used in continuous rotation (*Elnaghy & Elsaka, 2014*). It is made from M-Wire and shows better cyclic fatigue resistance than files made of conventional NiTi alloys (*Zupanc, Vahdat-Pajouh & Schäfer, 2018*) but lower torsional resistance than ProTaper Universal and ProTaper Gold (*Jamleh et al., 2021*). Shaping studies showed PTN files are able to respect the original canal shape (*Bürklein, Mathey & Schäfer, 2015*; *Wu et al., 2015*; *Hiran-us et al., 2016*; *Alrahabi & Alkady, 2017*; *Alshahrani & Al-Omari, 2019*). Compared to ProTaper Universal, PTN exerts lower vertical forces needed to negotiate canals (*Jamleh & Alfouzan, 2016*) and shows less canal transportation in severely curved canals (*Wu et al., 2015*; *Hiran-us et al., 2016*) and in simulated canals with double curvature (*Hiran-us et al., 2016*).

The Kerr Corporation recently introduced ZenFlex (ZF) (Kerr Corporation, Pomona, CA, USA) with a proprietary novel heat treatment. It has a triangular cross-section with a constant taper used in continuous motion. It has a maximum file diameter of 1 mm which allows to maintain more tooth structure after root canal therapy (https://www.kerrdental.com/kerr-endodontics/zenflex-shape#docs; last accessed 11 June 2023). Although ZF files exhibited cyclic fatigue resistance and bending properties similar to EdgeSequel Sapphire files, the former had improved torsional resistance (*Zanza et al., 2022*). ZF was also found to be stiffer and have reduced cyclic fatigue resistance and torsional resistance, compared to Vortex Blue files (*Zanza et al., 2022*).

It was shown that material type and ambient temperature influence the mechanical and physical properties of NiTi alloys which could have clinical implications (*Jamleh et al., 2016*; *Chien et al., 2022*). Testing the superelastic NiTi files at low temperature showed increased flexibility and resistance to cyclic fatigue (*Jamleh et al., 2016*). Heat-treated NiTi files exhibit different mechanical and physical properties at different temperatures (*Chien et al., 2022*).

Several shaping ability studies have been conducted to investigate the incidence of endodontic mishaps, including apical zipping, ledging, and file separation using various file systems with different designs, alloy microstructures, and shaping techniques in different canal configurations (*Bonaccorso et al., 2009*; *Bürklein & Schäfer, 2013*; *Shi et al., 2022a*; *Shi et al., 2022b*). However, the shaping ability of ZF has not been addressed. Furthermore,

most of the shaping ability studies have been performed at room or simulated body temperature (*Wu et al., 2015*; *Hiran-us et al., 2016*; *Alshahrani & Al-Omari, 2019*; *Gomaa, Osama & Badr, 2021*; *Orel et al., 2021*). Therefore, this investigation aims to determine the impact of cooling PTN and ZF files on their shaping ability in simulated canal models with double curvature at simulated body temperature and to compare between the two systems. The null hypothesis is file cooling does not affect the shaping ability of ZF files.

## MATERIALS AND METHODS

### Sample preparation

Endo-Training resin blocks with double curvature canals (Dentsply Sirona) were selected. The taper, diameter, and length of the canal are 0.02, 0.15 mm, and 16 mm, respectively. Coronally, the angle and radius of the curvature are 35 degrees and five mm, respectively. Apically, they are 30 degrees and 4.5 mm, respectively. A power analysis was conducted by selecting one-way independent ANOVA to determine the sample size needed using a significance level of 0.05. A sample size of at least 12 canals per subgroup was found to provide a statistical power of 0.90.

### Canal shaping

The working length (WL) was at the canal exit in all canals and a glide path was made with a ProGlider rotary file (Dentsply Sirona) to the WL. Then, the canal shaping was completed with PTN, cooled PTN, ZF, or cooled ZF ($n = 14$ each). In PTN, X1 (size 17, 0.04 taper) and X2 (size 25, 0.06 taper) files were employed in continuous rotation at 300 rpm. In ZF, two files (size 20, 0.04 taper and size 25, 0.06 taper) were employed in continuous rotation at 500 rpm. A random sequence was adopted in an attempt to eliminate bias toward the groups.

In the cooling groups, the file was subjected to refrigerant spray(Endo Frost) (Roeko, Langenau, Germany) continuously for 5 s at its cutting part just before being used for canal shaping.

The experiment was performed to simulate the clinical conditions where the simulated canal was immersed inside a warm water bath at simulated body temperature ($37 \pm 1$ °C) to the canal opening. The temperature was critically monitored with a thermocouple. The file was carefully inserted in the canals using a slow in-and-out picking motion and did not last longer than 5 s inside the canal. This was repeated until the file reached the WL. Between insertions, the file was cleaned and subjected to 5 s of cooling (in cooling groups). Each file was used to shape one canal and then discarded.

The canal was irrigated with a 1% NaOCl and recapitulation was made with a #10 K-file. Finally, the canal was irrigated with one mL of 17% EDTA.

The procedure was performed by an operator previously trained in both file systems using an X-Smart IQ device (Dentsply Sirona, York, PA, USA) with a maximum torque of 4 N. cm to measure the real-time torque during the procedure.

The shaping time, which was defined as the time taken by shaping and file changes, cleaning of the flutes, and irrigation of the canal and stopped when the sequence of the
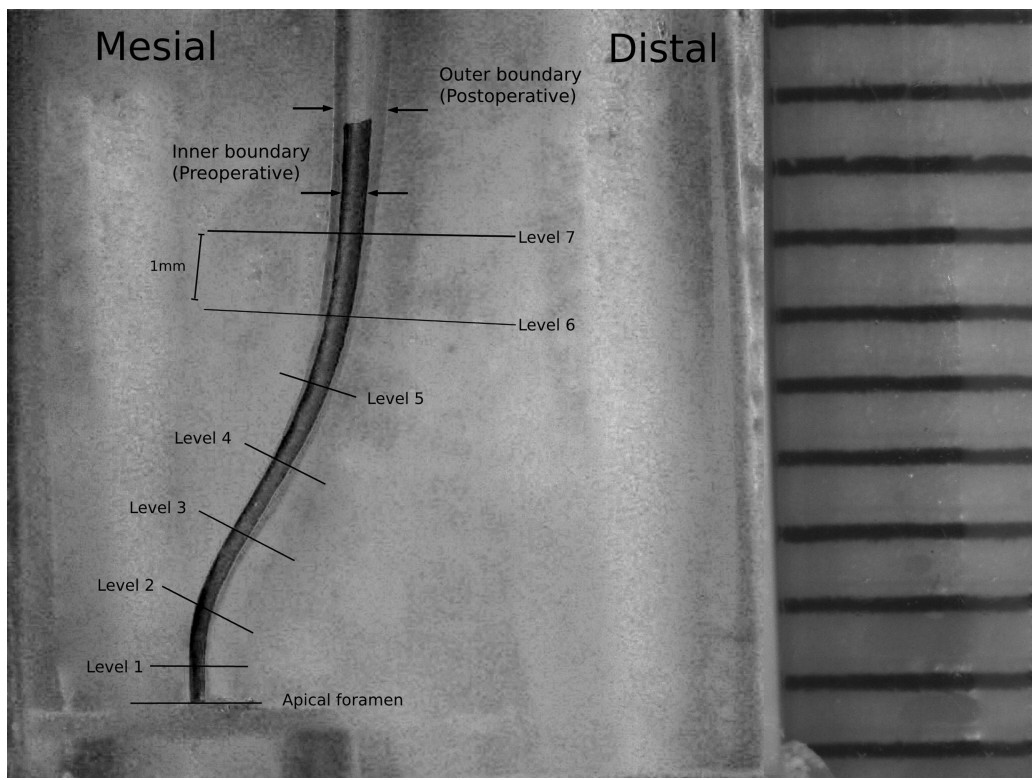

**Figure 1** Superimposition of pre- and post-operative images of a representative canal model with double curvature and localization of the measuring levels.

file is finished, was used for data analysis. The maximum torque value in each shaping was used for analysis.

## Assessment of canal shaping

Images for the canal models were taken before and after the shaping procedures by using a digital stereomicroscope (Leica EZ4 HD; Leica Microsystems, Singapore) and saved as tiff format files. Adobe Photoshop software was used to superimpose the pre- and post-operative images by accurately overlapping the reference points (Fig. 1). The amounts of removed resin from the mesial and distal walls were determined at eight measuring levels perpendicular to the surface of the canal wall surface. The first measuring level was the "canal exit" and the last measuring level was seven mm away from the canal exit. At each level, canal deviation was measured by subtracting the amount of removed resin on the mesial side from those on the distal side. All measurements were made by another operator who was blinded to the experimental groups.

## Assessment of file instruments

All files were examined with a microscope (OPMI pico; Carl Zeiss, Gottingen, Germany) at a magnification of 10 before and after each use to check for deformation or fracture.

**Table 1 The means and standard deviations of time and maximum torque measured during canal shaping in the experimental groups.**

| Group | Time (Seconds) | Maximum torque (N.cm) |
|---|---|---|
| **PTN** ($n = 14$) | $64.36 \pm 9.74$[a] | $1.69 \pm 0.19$ |
| **Cooled PTN** ($n = 14$) | $62.71 \pm 9.31$[a] | $1.61 \pm 0.44$ |
| **ZF** ($n = 14$) | $49.57 \pm 9.16$[b] | $1.59 \pm 0.27$ |
| **Cooled ZF** ($n = 14$) | $47.00 \pm 7.79$[b] | $1.63 \pm 0.29$ |
| **Statistical test** | ANOVA with LSD | Kruskal-Wallis |
| **$P$ value** | <0.05 | 0.54 |

Notes.
[a,b] Different superscript letters indicate statistical significance.
PTN, ProTaper Next; ZF, ZenFlex.

## Statistical analysis

The data were analyzed using the statistical package for social sciences (SPSS) software (Version 21; SPSS Inc, Chicago, IL, USA) at a 5% significance level. The Shapiro–Wilk test was used to determine the normality of the data. Shaping time was analyzed using ANOVA and LSD post-hoc tests. Differences in the maximum torque and canal deviation at each level were compared between the groups using the Kruskal–Wallis test.

## RESULTS

No file was deformed or fractured during the shaping of the canals with double curvature. All canals remained patent after shaping and were blocked with resin.

Table 1 shows the time and maximum torque attained during the shaping procedure. ZF and Cooled ZF required $49.57 \pm 9.16$ and $47.00 \pm 7.79$ s, respectively which were lower than PTN and Cooled PTN ($64.36 \pm 9.74$ and $62.71 \pm 9.31$ s) ($P < 0.05$; ANOVA with LSD tests). The maximum torque values were found comparable between the groups in the range from $1.59 \pm 0.27$ to $1.69 \pm 0.19$ N. cm ($P = 0.54$; Kruskal-Wallis test).

All the groups generated negligible deviations at every canal level evaluated which were comparable (Table 2) ($P > 0.05$; Kruskal–Wallis test). The canal deviation at the canal exit ranged from $-0.030$ to $0.001$ mm. The greatest mean canal deviation was found in PTN at the top coronal level with 0.100 mm toward the mesial side. The least canal deviations were found at the canal exit end and at a five mm level in the cooled ZF with 0.001 mm toward the distal side, and at a two mm level in the cooled PTN with 0.001 mm toward the mesial side.

There was a trend that the cooled PTN and ZF files exhibited lesser canal deviations than their counterparts. However, it was not statistically significant ($P > 0.05$).

## DISCUSSION

Since testing endodontic files at low temperatures has shown improved performance in cyclic fatigue and flexibility (*Jamleh et al., 2016*; *Shen et al., 2018*; *La Rosa et al., 2021*) investigating the shaping ability of cooled files has been suggested (*Jamleh et al., 2016*). The present study compared the ability of cooled PTN and ZF to shape simulated canals with

Jamleh et al. (2023), PeerJ, DOI 10.7717/peerj.15830

**Table 2  The means and standard deviations of canal deviation (mm) at different measurement levels.**

| Group | Measurement Level | | | | | | | |
|---|---|---|---|---|---|---|---|---|
| | 0 | 1 | 2 | 3 | 4 | 5 | 6 | 7 |
| PTN (n = 14) | −0.030 ± 0.052 | 0.023 ± 0.047 | 0.031 ± 0.079 | 0.078 ± 0.090 | 0.050 ± 0.098 | 0.003 ± 0.099 | −0.061 ± 0.089 | −0.100 ± 0.087 |
| Cooled PTN (n = 14) | −0.017 ± 0.050 | 0.025 ± 0.052 | −0.001 ± 0.137 | 0.047 ± 0.075 | 0.030 ± 0.088 | −0.008 ± 0.075 | −0.038 ± 0.073 | −0.084 ± 0.094 |
| ZF (n = 14) | −0.017 ± 0.065 | 0.005 ± 0.039 | −0.011 ± 0.035 | 0.003 ± 0.043 | −0.019 ± 0.047 | −0.040 ± 0.047 | −0.047 ± 0.048 | −0.056 ± 0.056 |
| Cooled ZF (n = 14) | 0.001 ± 0.070 | 0.017 ± 0.069 | 0.031 ± 0.093 | 0.022 ± 0.093 | 0.015 ± 0.098 | 0.001 ± 0.101 | −0.024 ± 0.105 | −0.047 ± 0.109 |

**Notes.**

Positive values represent canal deviation toward the distal side and negative values represent canal deviation toward the mesial side.

PTN, ProTaper Next; ZF, ZenFlex.

double curvature. At each level, negligible and comparable canal deviations were noticed in the tested groups. Thus, the null hypothesis was accepted.

ZF and cooled ZF were able to shape the canal quicker than PTN and cooled PTN while all groups had comparable maximum torque. Although two files were used to complete the shaping in each group, the shaping time was significantly different between groups. This might be attributed to the rotational speed, cross-sectional design, and heat treatment. The ZF is used at a higher rotational speed than PTN which makes the ZF shape the canals with 25% less time than the PTN. It is reported that with less shaping time there is a lower incidence for canal deviation to take place (*Hiran-us et al., 2016*). Also, the PTN has an off-centered design with decreasing taper makes the files rotate with a swaggering action that is claimed to facilitate canal space penetration and reduce canal transportation (*Bürklein, Mathey & Schäfer, 2015*; *Kyaw Moe et al., 2018*). This is also known to exhibit higher screw in force and ultimately an increase in the torque generated (*Glavičić et al., 2011*). Moreover, thermal treatment is known to influence cutting efficiency (*Pedullà et al., 2021*). Nonetheless, the tested file systems with different thermomechanical treatments were able to maintain the original canal geometry and the generated torque in all groups did not exceed 1.7 N. cm.

The file systems used in this study are made up of similar metal alloys but different thermal treatments. PTN is based on M-wire technology whereas ZF is based on proprietary heat treatment technology. The current findings of PTN groups are in line with the previous studies (*Bürklein, Mathey & Schäfer, 2015*; *Wu et al., 2015*; *Hiran-us et al., 2016*; *Alrahabi & Alkady, 2017*; *Alshahrani & Al-Omari, 2019*) in which the PTN exhibited less tendency of canal deviation at all measuring levels while respecting the original canal curvature. A previous study showed that PTN generated superior results at most levels compared with ProTaper Universal (*Hiran-us et al., 2016*). ZF system has been introduced recently with not enough previous shaping ability data to compare with the findings of this study.

Enlarging the canal preparation improves the cleaning efficiency through better canal debridement and flow of the irrigants into the apical space. Nonetheless, it may lead to undesirable canal mishaps, especially in narrow and curved root canals (*Fornari et al., 2010*). Therefore, it is suggested to shape canals with many curves with a NiTi file smaller than size 30 (*Bonaccorso et al., 2009*). Furthermore, a previous study revealed significant leakage in teeth with more than 0.3 mm apical transportation index (*Wu, Fan & Wesselink, 2000*). The apical deviation may complicate the obturation and result in a compromised apical seal (*Wu, Fan & Wesselink, 2000*). In this study, the canals were enlarged to a final apical size of 25, and the resultant shaping in the PTN and ZF groups did not deviate from the original foramen position of more than 0.229 mm.

The unique NiTi alloy properties have an impact on the performance of NiTi files represented by the amount of critical stress needed to induce austenite-martensite transformation. This transformation is greatly affected by applied stress and ambient temperature (*Daly, Ravichandran & Bhattacharya, 2007*). At an ambient temperature above the transformation temperature range, the NiTi alloy is composed mainly of austenite and exhibits significant hardness and stiffness values, whereas at a lower ambient temperature,

it consists mainly of martensite and exhibits reduced hardness and higher flexibility (*Daly, Ravichandran & Bhattacharya, 2007*; *Zhou, Peng & Zheng, 2013*).

After testing a series of application and shaping times for cooling the file to select a suitable regimen, each file was subjected to 5 s of continuous cooling with the Endo-Frost spray and then was used to shape the canal for no more than 5 s. Because the NiTi alloy has a very low specific heat (0.20 cal/g _C), it works as a quick and efficient heat absorber (*Tobushi et al., 1996*) that absorbs the energy developed during the NiTi phase transformation (*McKelvey & Ritchie, 2000*). Theoretically, reducing the ambient temperature below the austenite start temperature of the file will increase the percentage of martensitic phase in the alloy itself (*Shen et al., 2018*) which was shown to result in improved cyclic fatigue, flexibility, and bending properties leading to an extended lifetime of the NiTi files (*Jamleh et al., 2016*; *Grande et al., 2017*; *AbuMostafa & Alfadaghem, 2021*). This is also applicable to heat-treated files where the more martensitic configuration will take place (*Grande et al., 2017*; *AbuMostafa & Alfadaghem, 2021*; *Alfawaz et al., 2018*). The tested files received different thermal treatments. The austenite start and finish temperatures of PTN are 37.8 and 54.16 C, respectively, which are considered higher than the ZF (28.13 and 30.94 C, respectively) as shown in differential scanning calorimetry studies on file size 25 (*Zanza et al., 2022*; *Aminsobhani et al., 2016*). In this study, there was a trend that the cooled PTN and ZF files promoted lesser deviation from the original canal anatomy with double curvature than their counterparts.

None of the PTN or ZF files were fractured during shaping. A previous study showed that the PTN file system is safe to use in severely curved canals as none of the files were fractured during the canal shaping (*Alshahrani & Al-Omari, 2019*). Further studies are required to study the metallurgic properties of the tested cooled and as-received files.

Great care was made to purely investigate the effect of cooling on shaping ability within each tested system. Also, for a better standardization of the shaped samples, the last files used in PTN and ZF had a tip diameter equivalent to a size of 0.25 mm, and all files were used in a full rotary working motion to exclude any impact of the shaping size and different motion kinematics on the results.

Previous shaping studies of different NiTi files were conducted on natural teeth and simulated canals (*Hülsmann, Peters & Dummer, 2005*; *Peters, 2004*; *Kyaw Moe et al., 2018*; *Wu et al., 2015*; *Hiran-us et al., 2016*; *Alshahrani & Al-Omari, 2019*; *Gomaa, Osama & Badr, 2021*; *Orel et al., 2021*; *Alfadley et al., 2020*). It is known that high variability in root canal anatomy is present which may act as a confounder affecting the shaping results (*Peters, 2004*). Although resin blocks lack qualities of human dentin which calls for caution to extrapolate the current findings to real clinical conditions (*Hülsmann, Peters & Dummer, 2005*; *Peters, 2004*), the use of resin blocks with simulated canals has been validated by other shaping studies (*Shi et al., 2022a*; *Shi et al., 2022b*). The use of resin models with similar canal diameter, length, and curvature in three dimensions is highly favorable. Moreover, image superposition is an accepted technique to assess canal shaping efficiency (*Bonaccorso et al., 2009*). Thus, for better standardization, the current findings were obtained from resin canal models with double curvature to allow for comparing the shaping performance of two thermally treated NiTi files.

This is the first *in vitro* study to test the effect of cooling on file's shaping ability which could be considered clinically feasible as proof of concept. The currently tested parameters were chosen to answer the study question and present the advantages and disadvantages of the assessed technique. However, a more comprehensive assessment should be considered by adopting a multi-method approach (*Martins et al., 2022*) to maximize understanding of the mechanical performance of endodontic files and generalize the study outcomes.

# CONCLUSIONS

Under the conditions of this study, the tested file systems demonstrated similar shaping ability whilst maintaining the original curvature of the canal in simulated canals with double curvature. There was a trend that cooled files made lesser canal deviations compared to their counterparts.

## Funding
This study was supported by the King Abdullah International Medical Research Center (No. NRC22R-579-11). The funders had no role in study design, data collection and analysis, decision to publish, or preparation of the manuscript.

## Grant Disclosures
The following grant information was disclosed by the authors:
King Abdullah International Medical Research Center: NRC22R-579-11.

## Competing Interests
Nawaf Alshetan and Sulaiman Alraffa are employed by Dr. Sulaiman Al-Habib Medical Group

## Author Contributions
- Ahmed Jamleh conceived and designed the experiments, performed the experiments, analyzed the data, prepared figures and/or tables, authored or reviewed drafts of the article, and approved the final draft.
- Hajar Albanyan performed the experiments, authored or reviewed drafts of the article, and approved the final draft.
- Ali Alaqla performed the experiments, authored or reviewed drafts of the article, and approved the final draft.
- Hamad Alissa conceived and designed the experiments, performed the experiments, authored or reviewed drafts of the article, and approved the final draft.
- Nawaf Alshetan performed the experiments, analyzed the data, authored or reviewed drafts of the article, and approved the final draft.
- Sulaiman Alraffa performed the experiments, authored or reviewed drafts of the article, and approved the final draft.
- Abdulmohsen Alfadley conceived and designed the experiments, performed the experiments, authored or reviewed drafts of the article, and approved the final draft.

## Data Availability

The raw data is available in the Supplemental Table.

## Supplemental Information

Supplemental information for this article can be found online at http://dx.doi.org/10.7717/peerj.15830#supplemental-information.

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
