# Peer review of "Impact of cooling on shaping ability of thermally treated files in canal models with double curvature"

_PeerJ, doi:10.7717/peerj.15830_

## Round 0.1 · original submission · Major Revisions

Kindly consider implementing the reviewer's suggestions in your manuscript.

Reviewer 1 ·

Basic reporting

Revision of the manuscript entitled “Impact of cooling on shaping ability of thermally treated files in canal models with double curvature” submitted to PeerJ journal.

The present study addresses the impact of cooling the instruments on their shaping ability. I have a few concerns that I would like to express to the authors.

Experimental design

The introduction is globally ok. However, the importance given to the cooling issues is not properly addressed. If this is one of the main characteristics and therefore more relevance should be given to this topic. I suggest the authors to use the manuscript DOI 10.3390/ma15238367 to debate the topic and make a study rational since they have tested mechanically 4 different raw NiTi wires at 3 different temperature (including a cooled one).

As a rationale for the study, the sentence “Moreover, the effect of low temperature upon the NiTi file.s shaping ability has not been addressed” is too limited and more arguments should be presented.

Regarding the aim sentence, the authors have also compared instruments between them, this should be stated as a secondary goal.

The option “In the cooling groups, the file was subjected to refrigerant spray (Endo Frost) (Roeko, Langenau, Germany) continuously for 5 seconds at its cutting part just before being used for canal shaping” should be debated in the Discussion and the authors should try to search in the literature of how much this can interfere in the crystallographic arrangement of the instruments? Is this enough to make them full martensitic?

Regarding the option “The experiment was performed to simulate the clinical conditions where the simulated canal was immersed inside a warm water bath at simulated body temperature (37±1 °C) to the canal opening”. If the experiment (cyclic fatigue) does not mimic a clinical condition and is the goal was to test the instruments cooled… why did your warmed later? This option should be explained in the manuscript and how much this can interfere with the cooling assessment.

Why an n=14 was used? Have you conducted any sample size determination?

Change the sentence “However, it was not statistically significant.” to “However, it was not statistically significant (P>0.05).”

Validity of the findings

I really feel that all this cooling and heating the instruments may not give reliable answers because the authors have basically no clue at which temperature the instruments were during the test itself. The temperature was not controlled effectively. The authors try to conduct the experiments in a way that the relevant variables are indeed controlled. I believe this issue should be debated in the limitations of the study. Moreover, to conducted a comprehensive assessment a multi-method approach should be done as appears to be the tendency currently (DOI 10.1016/j.joen.2022.05.007), which I also suggest to be mentioned.

Strength of the study should be debated.

Generalizations of the study outcomes should be also debated.

·

Basic reporting

The use of English in this manuscript was clear and concise. The article is well structured with appropriate updated references.

Experimental design

I have some concerns in the experimental design, which must be elaborated to improve the quality of the paper.

Line 86 - PTN X1 and X2, please outline the ISO dimensions, as in Line 87 the ZF dimensions are given. Are the ISO dimensions comparable?
Line 86 - As well, why is n=14? Was a power analysis done beforehand to arrive at this sample? No reason was given
Line 86-87 what are the manufacturer's settings for PTN and ZF? Are they the same (I don't believe so, as ZF has very specific settings). How were the torque and RPM justified
Line 87-88 "A random sequence was adopted" how was this randomised?
Line 97-98 The irrigation procedure is interesting, as NaOCl and EDTA are not required realistically for resin blocks. Were there any potential interactions? Why were those irrigants used over distiled water or saline?
Line 102: The shaping time was "defined as the time taken by shaping and file changes, cleaning of the flutes, and irrigation of the canal and stopped when the sequence of the file is finished" How can this be standardised if the cleaning of flutes and irrigation is counted? Was the cooling time taking into account? Why is shaping time not when limited to when the instruments are in the canal? This is a critical point that must be addressed.
Lines 115-116 Perhaps a subjective analysis between cooled and uncooled files may have been interesting.

Validity of the findings

Line 156: I understand ZF heat treatment if proprietary, but how are the effects of temperature accounted for if the type of treatment is unknown ? In Line 172 the authors correctly say that ambient temperature affects the austenite-martensite transformation. But without appreciation of the austenite-martensite relationship in both wires, we cannot define the impact of the effect. This is a critical limitation.

Are less canal deviations purely a result of cooling or other possible confounding variables?

Additional comments

The authors have executed some honest experimentation with interesting results, but some processed in the methodology must be explained in more detail and/or justified, as this impacts the validity of the findings.

---

## Round 0.2 · accepted · Accept

I appreciate you for submitting your article to PeerJ.

Reviewer 1 ·

Basic reporting

Dear authors, I have no more concerns regarding the basic reporting.

Experimental design

Dear authors, I have no more concerns regarding the experimental design.

Validity of the findings

Dear authors, I have no more concerns regarding the validity of the findings.

Additional comments

Dear authors, I have no more additional comments.